# Effects of transcranial direct current stimulation (tDCS) and concurrent cognitive training on episodic memory in patients with traumatic brain injury: a double-blind, randomised, placebo-controlled study

Daglie Jorge De Freitas [1], Daniel De Carvalho,[1] Vanessa Maria Paglioni,[1] Andre R Brunoni [2,3] Leandro Valiengo [2,3] Maria Sigride Thome-Souza,[2] Vinícius M P Guirado [1] Ana Luiza Zaninotto [1,4] Wellingson S Paiva [1]

ALZ and WSP are joint senior authors.

For numbered affiliations see end of article.

**Correspondence to**
Dr Ana Luiza Zaninotto;
acostazaninotto@mghihp.edu

## ABSTRACT

**Introduction** Deficits in episodic memory following traumatic brain injury (TBI) are common and affect independence in activities of daily living. Transcranial direct current stimulation (tDCS) and concurrent cognitive training may contribute to improve episodic memory in patients with TBI. Although previous studies have shown the potential of tDCS to improve cognition, the benefits of the tDCS applied simultaneously to cognitive training in participants with neurological disorders are inconsistent. This study aims to (1) investigate whether active tDCS combined with computer-assisted cognitive training enhances episodic memory compared with sham tDCS; (2) compare the differences between active tDCS applied over the left dorsolateral prefrontal cortex (lDLPFC) and bilateral temporal cortex (BTC) on episodic memory and; (3) investigate inter and intragroup changes on cortical activity measured by quantitative electroencephalogram (qEEG).

**Methods and analysis** A randomised, parallel-group, double-blind placebo-controlled study is conducted. Thirty-six participants with chronic, moderate and severe closed TBI are being recruited and randomised into three groups (1:1:1) based on the placement of tDCS sponges and electrode activation (active or sham). TDCS is applied for 10 consecutive days for 20 min, combined with a computer-based cognitive training. Cognitive scores and qEEG are collected at baseline, on the last day of the stimulation session, and 3 months after the last tDCS session. We hypothesise that (1) the active tDCS group will improve episodic memory scores compared with the sham group; (2) differences on episodic memory scores will be shown between active BTC and lDLPFC and; (3) there will be significant delta reduction and an increase in alpha waves close to the location of the active electrodes compared with the sham group.

**Ethics and dissemination** This study was approved by Hospital das Clínicas, University of São Paulo Ethical Institutional Review Border (CAAE: 87954518.0.0000.0068).

## Strengths and limitations of this study

► To our knowledge, this protocol is the first randomised controlled trial investigating the effects of the transcranial direct current stimulation (tDCS) and concurrent computer-assisted cognitive training on episodic memory in individuals with sustained TBI.

► This study may contribute to the development of evidence-based low-risk and low-cost rehabilitation treatment for individuals with TBI and memory impairments.

► We will compare differences on episodic memory outcomes based on the anodal tDCS electrode placement in the cortex.

► Electroencephalogram will be used to evaluate changes in cortical activity after the intervention.

► Due to sample size restrictions, sex and TBI severity will not be considered as covariates, which might be a limitation of this study.

**Trial registration number** NCT04540783.

## INTRODUCTION

Traumatic brain injury (TBI) is an alteration in brain function caused by an external force and a major cause of death and disability throughout the world.[1 2] The hippocampus and the prefrontal cortex are among the brain structures more susceptible to lesions after a brain insult and, as a consequence, head injury survivors may experience difficulties in recalling specific events from the personal past and imagining novel scenarios.[3–5] Those regions are known to play important roles in episodic memory, which is a declarative

memory containing information about place and time of acquisition as opposed to semantic memory, which refers to memory not tied to the context of encoding.[6] The hippocampus specifically organises experienced and biographical memories that are defining features of episodic memory, and the prefrontal cortex suppresses context-inappropriate memories thus allowing the retrieval of context-appropriate memories.[7] After brain trauma, cognitive impairment might be persistent[8] and no available treatments have been shown to be effective to improve those sequels.

Non-invasive brain stimulation (NIBS) techniques, including transcranial direct current stimulation (tDCS), are neuromodulatory interventions that have been shown to improve neuroplasticity and cognitive outcomes in neurological conditions, including TBI.[9 10] tDCS can transiently alter neuronal activity facilitating or inhibiting neuronal circuitries depending on the polarity of the stimulation.[11] tDCS induces neuroplasticity by applying a low-intensity electrical current (0.5–2 mA) through electrodes placed on the scalp. The electrodes have two polarities (anode and cathode) and change the resting state of the membrane cells of the surrounding region.[12 13] Previous studies have shown that repetitive tDCS sessions improved disorder of consciousness[11 14 15] and cognition in patients with TBI,[8 16] whereas some studies have shown inconsistent results.[17–19]

Since tDCS works on the membrane level, changing the resting state but not evoking action potential, the use of tDCS with concurrent cognitive training seems to be a good option to potentiate the stimulation and modulate the brain networks accordingly to the target training.[20–23] Two prior studies investigated the effects of the use of tDCS and cognitive training (non-concurrent) on memory and attention performance in TBI patients, but only one found a significant improvement in the cognitive outcome measures.[20 22]

Biomarkers that evaluate brain changes after the tDCS intervention are still scarce, however, the electroencephalogram (EEG) has been suggested to be a useful tool for this purpose.[24–29] The EEG measures the rhythm of electrical activity in the brain according to its frequency: delta (1–4 Hz), theta (4–8 Hz), alpha (8–12 Hz), beta (12–30 Hz) and gamma (30–40 Hz)[27 30 31] and is widely used as a safety outcome in patients who undergo tDCS sessions.[32–35] Some studies associate EEG measures (amplitude, power, phase and coherence) to the functionality of patients,[36] including the diagnosis and prognosis of patients with TBI.[37 38] A study using EEG power spectrum[39] suggests that, after 10 tDCS sessions, changes in the rhythm of brain activity occur, with reduction of delta and increase of alpha near the active electrodes in patients with chronic TBI. This study also found a significant correlation between decreases in delta and improved performance on neuropsychological tests for the active tDCS group to far greater extent than for the sham group.[24] Other studies have measured cortical activity after a single session of tDCS and have shown

inconsistent results.[24–29] Thus, cortical changes after consecutive sessions of tDCS combined with cognitive training in people with TBI are still inconclusive.

Due to the lack of consensus and scarcity of evidence about the effects of cognitive training in addition to tDCS sessions in patients with TBI, the goals of this study are (1) to investigate the effect of 10 sessions of tDCS and concurrent cognitive training in patients with TBI compared with sham tDCS; (2) to analyse differences on episodic memory scores between active anodal tDCS over the left dorsolateral prefrontal cortex (lDLPF) and bilateral temporal cortex (BTC) and (3) to analyse changes on cortical activities (measured by the EEG) between the groups. We hypothesise that (1) participants that received active stimulation will have greater scores on episodic memory test compared with the sham group; (2) there might be significant score differences on episodic memory test between patients who were stimulated over the BTC and those stimulated over the lDPFC and (3) delta reduction and an increase in alpha waves close to the sponge placement in the active group compared with the sham group.

## METHODS AND ANALYSIS
### Design
This is a randomised, parallel-group, placebo-controlled and double-blind study that is being conducted at Hospital das Clínicas, Faculdade de Medicina da Universidade de São Paulo, (HC-FMUSP), São Paulo, Brazil. Participants who meet eligibility criteria are randomly allocated to (1) group 1—BTC; (2) group 2—lDLPFC and (3) group 3—sham (BTC or lDLPFC).

A 20 min-tDCS for 10 days (2 weeks, except for Saturdays and Sundays) is delivered simultaneously to a computer-assisted cognitive training (for 20 minutes). Patients will be assessed at baseline (T0), at the end of the last stimulation session (T1) and 3 months after the last tDCS session (T2) (figure 1).

### Ethics committee and regulatory approval
The trial is conducted in accordance with the ethical principles outlined in the Declaration of Helsinki, 1996. This research was approved by the Hospital das Clínicas, University of São Paulo Ethical Institutional Review Border number CAAE: 87954518.0.0000.0068. Any severe side effect during the trial will be reported to the safety monitoring board IRB for appropriate management.

### Randomisation and blinding
The investigator ALZ was responsible for the computer-generated random assignment list, arranging patients in blocks of 3 or 6. The proportion of the randomisation for each group is 1:1:1. This randomised list ensures double blinding so that both research assistants and patients are blind to the type of stimulation. Before each stimulation session, the researcher responsible for the stimulation receives a code that allows the tDCS device to deliver

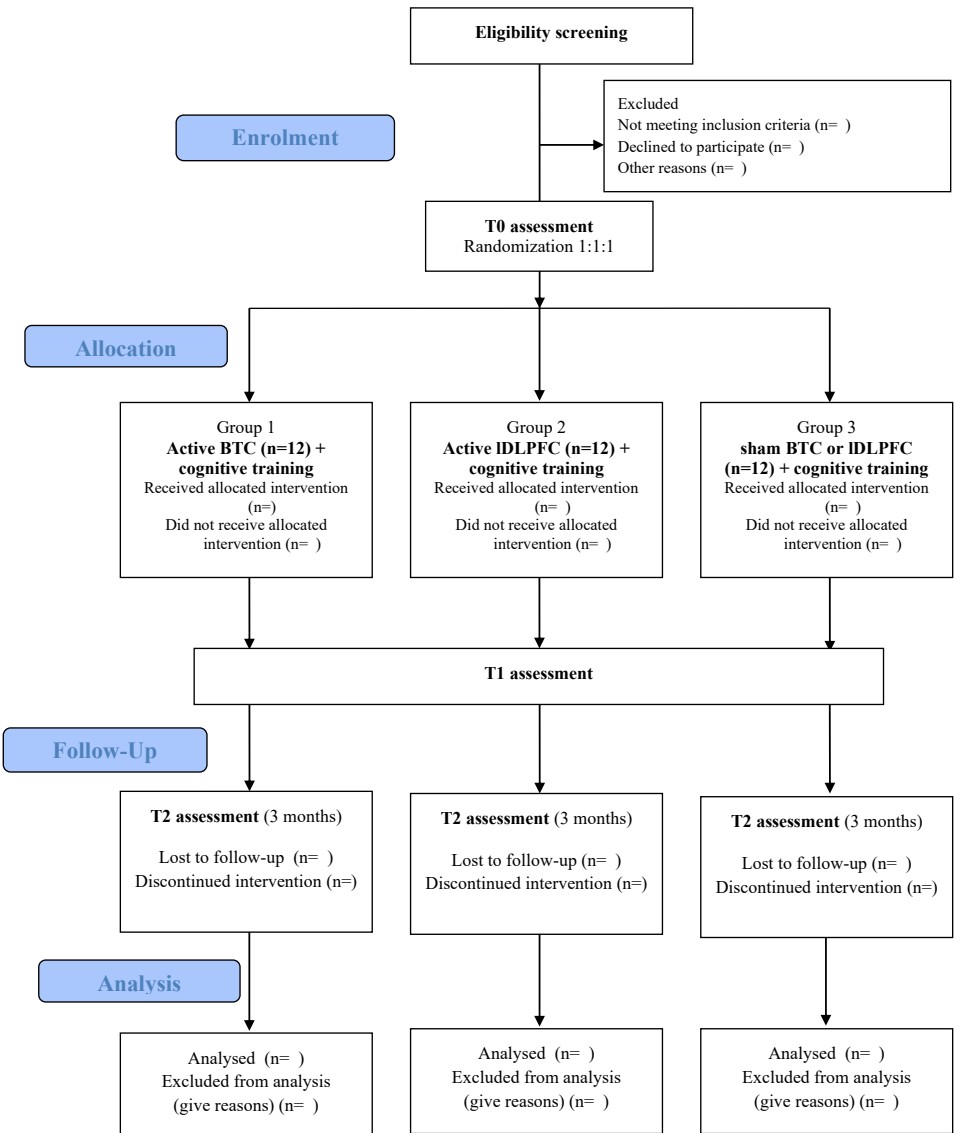

**Figure 1** CONSORT flow diagram. BTC, bilateral temporal cortex; CONSORT, Consolidated Standards of Reporting Trials; DLPFC, dorsolateral prefrontal cortex.

20 min of active or sham stimulation. This blinding and methodological procedure is similar to the rational of previous studies.[20 22 24]

The randomisation list and the NeuroConn (tDCS device) code is kept inside a locked drawer with restricted access at the research coordination office at HC-FMUSP.

### Recruitment and study population

Thirty-six patients with TBI are being recruited from hospitals in São Paulo. All participants provide written informed consent and receive an exclusive identification number during the screening period, to ensure blinding. Study recruitment started in June 2019 and the estimated completion date for the primary outcome is June 2022. We expect that 85% of the patients will be inpatients from HC-FMUSP referred by neurologists and 15% from extramural recruitment (from social media and folders). This

trial follows the Consolidated Standards of Reporting Trials guidelines.

### Inclusion criteria

► Outpatients with radiological diagnosis of TBI at least 6 months prior to enrolment in the study.
► Glasgow Coma Scale score ≤12 at admission in the emergency room.
► Memory complaints, self-reported or reported by the family/caregiver.
► Age between 18 and 55 years.
► Able to follow directions.

### Exclusion criteria

► History of epilepsy post-TBI.
► Clinical EEG abnormalities (epileptiform activity, disorganised background, in other words, a general

| Table 1 | Detail of the study visits | | | | |
|---|---|---|---|---|---|
| | Eligibility screening tasks | Visit 1 baseline | Visit 2–9 | Visit 10 | Visit 11, 3 months follow-up |
| Consent form | X | | | | |
| Medical history | X | | | | |
| qEEG | X | | | X | X |
| BDI-II | X | | | X | X |
| BAI | X | | | X | X |
| Estimated IQ | X | | | | |
| RAVLT | | X | | X | X |
| tDCS | | X | X | X | |
| Cognitive Training | | X | X | X | |
| AEQ | | X | X | X | |

AEQ, Adverse Events Questionnaire; BAI, Beck Anxiety Inventory; BDI-II, Beck Depression Inventory; qEEG, quantitative electroencephalogram; RAVLT, Rey Auditory Verbal Learning Test; tDCS, transcranial direct current stimulation.

change in the way a normal brain wave looks—frequency, height and shape).

► Uncorrected visual impairment.
► Contraindications to tDCS, such as medical devices implanted in the brain or metallic foreign body in the head.
► Current severe/major depression (score over 36 points on the Beck Depression Inventory-second edition (BDI-II)).
► Current severe anxiety (score over 26 points on the Back Anxiety Inventory (BAI)).
► Limiting motor deficit.
► Estimated IQ under 70.
► Time after trauma >18 months

## Patient and public involvement

Patients are not involved in the design, or conduct, or reporting, or dissemination plans of our research.

## Instruments

Patients are expected to come to the research hospital for 11 visits as described in table 1.

## Screening assessment

Depressive symptoms—BDI-II.[40]

Anxiety symptoms—BAI.[41]

Estimated IQ—Wechsler Adult Intelligence Scale (Matrix Reasoning and Vocabulary).[42 43]

## Primary outcome (episodic memory)

Rey Auditory Verbal Learning Test (RAVLT): A list of 15 words is presented, and individuals are asked to recall as many words as possible after each trial, five in total (trials A1–A5). Twenty minutes after the fifth trial, a different list of 15 words (list B) is presented, followed by a free-recall test. A delayed recall of the first list (trial A6) and after 20 minutes delay (trial A7) are performed, and the individuals are requested to recall as many words as possible.[44] The seventh trial of the list A (trial A7) will

be used as our primary outcome. Different and random versions of the RAVLT (adapted for Brazilian Portuguese) are used to avoid learning bias between the assessments time points (T0, T1, and T2).

## Secondary outcome

EEG assessment: The examination is performed on the Nihon Kohden EEG 1200 V.01.71 digital equipment, with simultaneous video recording with a Sony Ipela camera. For qualitative EEG data analyses for abnormal spikes, we use the international 10–20 electrode placement system, 19 channels (being one to ECG), with sampling rate of 200 Hz, a time of 0.3, high filter from 35 to 70 Hz and sensitivity of 7 µV. For the quantitative analysis, the data are converted using Neuromap from the Neurowork-bench software. The examination lasts 30 min (15 min with your eyes open and 15 min with your eyes closed—relaxed wakefulness). The analyses are performed by a certified neurophysiologist (MSTS).

## Safety screening

Adverse Events Questionnaire (AEQ): Questionnaire that must be answered after each stimulation session to assess adverse effects such as tingling sensations, itching, mild transient redness of the skin and discomfort on the region of stimulation, moderate fatigue, difficulty concentrating, headache and nauseas.[45]

## Intervention

### Transcranial direct current stimulation

Both anodal and sham tDCS will be delivered by the same battery-driven (neuroConn: DC Stimulator Plus), for 20 min. The research assistant will set up the device according to the assignment list in order of participant's registration number. Saline-soaked surface 35 cm² (5×7 cm) sponge with electrodes connected to the stimulator will be placed on the patient's scalp and secured with adjustable rubber straps.

The sponge placement follows the 10–20 EEG system. Group 1—BTC—two anode electrodes are placed over T3 and T4, respectively, and the cathode electrode over the supraorbital region (FP2). Group 2—lDLPFC, the anode electrode is placed over F3 and the cathode over FP2. Group 3—sham group—half of the participants are following the montage of group 1 (BTC) and the other half from group 2 (lDLPFC). T3, T4 and F3 regions have been chosen for this protocol because other studies have investigated the effects of tDCS on memory by placing the electrodes over those regions.[46–55] For the sham stimulation, patients receive the active current with ramping up and down for 30 s to simulate the real stimulation over the BTC or lDLPFC, as referred by other studies.[56 57] Patients are monitored daily for side effects, according to international safety guidelines, and with the AEQ.[45]

## Cognitive training

The Rehacom is a cognitive software for patients with different aetiologies approved by the Brazilian Health Regulatory Agency. This software has several cognitive modules. For the purpose of the present study, we are using the attentional visual and verbal memory training tasks with increasing levels of difficulty according to the patient's performance. During the training, the feedback option is active, so the patient can be oriented and improve his/her performance over the trials. The initial level is adjusted to level 1 for all patients who have incomplete high school, for those who complete high school the starting level is 4, and for those with complete college, the starting level is 5.[58–60]

The cognitive training follows two possible random sequence order—memory/attention or attention/memory modules, always alternating daily up to the end of the last stimulation session. Each training has a 20 min duration, always combined with the tDCS.

## Sample size calculation

The sample calculation was performed using the software GPower V.3.1, using the statistical two-way analysis of variance (ANOVA) (three groups and three time points), a given α 5%, power 80% and interaction effect of 0.25 considering the primary outcome, based on our pilot data. G power analysis provided a sample size of 36 participants based on the F calculation (12 patients per group).

## Statistical analysis

Descriptive statistics are used to report demographic data. Kolmogorov-Smirnov test was used to test data normality. To analyse the primary outcome, changes on episodic memory (RAVLT scores of the 7th trial - A7), we will use ANOVA for normal data distribution or non-parametric tests. We assume that each participant has a random effect on the model. For the secondary outcome (EEG spectral power), we plan to use the mixed effect model (reml), considering group and time as fixed factors and each participant as a random effect. Estimated alpha value of 5%. An intention-to-treat framework will be applied.

## Ethics and dissemination

TDCS is a safe intervention not only because the electric current applied is very low (0.5–2 mA over a 25–35 cm² area), but also because the electrodes embedded in saline solution minimise tissue resistance, avoid overheating. Tingling sensations, itching, mild transient redness of the skin and discomfort on the region of stimulation, moderate fatigue, difficulty concentrating, headache and nauseas are possible adverse effects, but these effects do not usually last long and are often seen at the same frequency in experimental and placebo groups.[10 61 62] The Safety Side Effect Questionnaire (AEQ[45]) is collected after each stimulation session. Complaints regarding the stimulation or high AEQ scores are reported to the safety board and the medical coverage may be called for necessary care. Written informed consent for participation in the study will be obtained from all participants. Participant information is stored securely in locked file cabinets and participant digital information is password protected.

## DISCUSSION

In order to contribute to the development of evidence-based rehabilitation treatments to TBI survivors with memory impairments, the present study aims to investigate whether the use of tDCS targeting the BTC or the lDLFFC with concurrent computer-based cognitive training improves memory performance in patients with moderate and severe closed TBI.

Since TBI causes health loss and disability for individuals and their families[63] and memory impairment is one of the most frequent cognitive complaints,[64 65] an effective rehabilitation tool will be helpful to improve this burden in this population.

There is evidence that tDCS may improve cognitive impairments, such as memory impairments, following TBI and other aetiologies.[49 50] Prior research has shown the efficacy of anodal tDCS in improving memory performance during tasks such as face-name associative recall tasks, intentional memoriszation of words, figure-naming tests, word recall and picture-pseudoword associative learning tasks.[51–55] A recent systematic review found 14 experimental studies on adult patients with TBI who received tDCS for the assessment of clinical or surrogate outcomes[66] and, to our knowledge, only two studies used tDCS concomitant to cognitive training (non-concurrent) in patients with TBI.[20–22]

Despite some disadvantages, namely poor spatial/temporal resolution and stimulation of large part of the brain, there are many advantages to tDCS, such as low risk of adverse effects and low cost.[67] It has been proven that tDCS does not induce depolarisation, meaning it does not induce the firing of neurons when they are not near threshold. Therefore, it is less likely that neurons not engaged in the task at hand will discharge, hence the importance of applying tDCS during a specific task in order to target a particular circuitry.[23] It has also been suggested that more systematic investigations are

needed, due to the heterogeneity of findings in tDCS research and the different parameters used in the stimulation.[52 68]

EEG will be used to guarantee safety[32–35] and to measure cortical activity post intervention. Spikes and abnormal waves shown on the EEG will provide clinical guidance on whether to include the participant in the present study. We expect to reduce delta activities and increase alpha frequencies close to the active electrodes and find a better performance correlation in neuropsychological tests in the active group, as demonstrated previously.[24]

One limitation of this study is that, due to sample size restrictions, sex and TBI severity will not be considered as covariates. Severe TBI and moderate TBI are considered as a single entity for investigation purposes in many studies, in part because of the permanent physical, cognitive and behavioural impairments that are observed in such patients in comparison to patients with mild TBI. As for sex differences, a recent study aimed at characterising the demographic, social and economic profile of patients with TBI in Brazil showed that men were hospitalised almost 3.5 times more frequently for TBI than women and that the incidence of TBI in the male population was 102/100 000/year.[69–71]

This is a study to test the effectiveness of combined tDCS and cognitive training to improve episodic memory in patients with TBI. The results generated may potentially be useful for other neurological disorders that cause cognitive impairments. Our open-label pilot study (n=4 participants) has proven the feasibility of the method and a moderate effect size of the RAVLT scores between the baseline to the last tDCS session. Results will be presented at conferences and submitted for publication in peer-reviewed journals.

## Trial status

The open-label pilot study was performed with four participants in 2018 and validated the study protocol. Recruitment started in February 2019. At the time of submission of this paper, we had completed 15 participants. The programmed completion date for the primary outcome is June 2022.

This study will provide important data regarding the use of the combined tools to improve the memory of persons that suffer from the sequela of a TBI. Larger clinical trial studies are required to further interrogate the clinical efficacy of this technique to improve the mood and the quality of life of this target population.

## Author affiliations

¹Division of Neurology/Neurosurgery, Hospital das Clinicas, Faculdade de Medicina da Universidade de Sao Paulo, HCFMUSP, Sao Paulo, Brazil
²Institute of Psychiatry, Hospital das Clinicas da Universidade de Sao Paulo, IPq HCFMUSP, University of São Paulo, São Paulo, Brazil
³Interdisciplinary Center for Applied Neuromodulation and Service of Interdisciplinary Neuromodulation, University of Sao Paulo, Sao Paulo, Brazil
⁴Speech and Feeding Disorders Lab, MGH Institute of Health Professions, Boston, Massachusetts, USA

**Acknowledgements** The authors thank the Institute of Psychiatry, Hospital das Clínicas (IPq HCFMUSP), the secretary Sandra Aparecida de Lima Falcon and all participants and their families for their support and assistance.

**Contributors** Research project: Conception: ALZ and WSP. Organisation: ALZ, WSP, ARB, LV and VMPG. Execution: DJDF, VMP, DDC and MSTS. Statistical analysis: Design: ALZ, WSP and ARB. Manuscript Preparation: first draft: DJDF. Final draft: DJDF, VMP, DDC and ALZ. Review and critique: ALZ, ARB, LV, VMPG, DDC, DJDF and VMP.

**Funding** VMP reports grant from coordination for the Improvement of Higher Education Personnel (CAPES)—Brazil. ALZ is supported by the National Institute of Health NIDCD 3R01DC017291-02. WSP is supported by the NIHR Global Health Research Group on Neurotrauma, which was commissioned by the National Institute for Health Research (NIHR) using UK aid from the UK Government (project 16/137/105).

**Disclaimer** The views expressed in this publication are those of the author(s) and not necessarily those of the NIH, NIDCD, NIHR or the Department of Health and Social Care.

**Competing interests** None declared.

**Patient and public involvement** Patients and/or the public were not involved in the design, or conduct, or reporting, or dissemination plans of this research.

**Patient consent for publication** Not required.

**Provenance and peer review** Not commissioned; externally peer reviewed.

**ORCID iDs**
Daglie Jorge De Freitas http://orcid.org/0000-0002-2473-6210
Andre R Brunoni http://orcid.org/0000-0002-6310-3571
Leandro Valiengo http://orcid.org/0000-0002-7628-9993
Vinícius M P Guirado http://orcid.org/0000-0002-7782-3609
Ana Luiza Zaninotto http://orcid.org/0000-0003-0639-0612
Wellingson S Paiva http://orcid.org/0000-0002-5165-7040

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
