## [Reviewer comments · BMJ Open]

ARTICLE DETAILS

TITLE (PROVISIONAL)	The effects of transcranial direct current stimulation (tDCS) and concurrent cognitive training on episodic memory in patients with traumatic brain injury: a double-blind, randomized, placebo-controlled study
AUTHORS	De Freitas, Daglie; De Carvalho, Daniel; Paglioni, Vanessa Maria; Brunoni, Andre; Valiengo, Leandro; Souza, Maria Sigride; Guirado, Vinicius; Zaninotto, Ana Luiza; Paiva, Wellingson

VERSION 1 – REVIEW

REVIEWER	Cheng, Shih-kuen National Central University, Institute of Cognitive Neuroscience
REVIEW RETURNED	05-Dec-2020

GENERAL COMMENTS	This study describes a protocol to examine whether combining tDCS (over the left DLPFC and the bilateral temporal cortex, BTC) and cognitive training (attention and verbal memory) over 10 sessions enhance the episodic memory function of traumatic brain injured (TBI) patients. The Rey Auditory Verbal Learning Test (RAVLT) will be used to index episodic memory function. The changes in the power of the alpha and delta bands will also be used to index the effect of tDCS. The aim of this paper is clearly stated in the paper. The protocol is in general properly described. There are however a few things that the authors might want to consider regrading this research. 1. The cognitive function this study wanted to focus is episodic memory. As also noted by the author, episodic memory is related to the memory for the spatial and temporal context of experienced events. How is this nature of episodic memory (spatial and temporal context) indexed by the RAVLT? More specifically, what scores obtained in the RAVLT would be used to index episodic memory?2. As also noted by the authors in the introduction, episodic memory is subserved by the hippocampus and the left DLPFC (along with a number of other brain regions). It is therefore understandable that the left DLPFC is chosen as the location for the tDCS, but why was the bilateral temporal cortex also selected as the target sites? It is also not clear why it is predicted that stimulating the BTC would have a greater episodic memory enhancement than stimulating the left DLPFC? If indeed such difference is observed, would that a result of different stimulation magnitude over the DLPFC and the hippocampus?3. The RAVLT will be tested three times in the protocol (Visit 1 as the baseline, visit 10, and visit 11). Will the same lists be used in the
---

	three tests? 4. The protocol did not make it clear the temporal relationship between the tDCS stimulation and the cognitive training. 5. The sample size was estimated based on the 3 (DLPFC, BTC, and sham stimulation) by 3 (testing times) interaction effect on the episodic memory performance with an effect size of .25. What was the basis for the effect size of .25?
--	---

REVIEWER	Bragin, Denis University of New Mexico School of Medicine, Neurosurgery
REVIEW RETURNED	03-Feb-2021

GENERAL COMMENTS	The authors present the plan for a randomized controlled trial investigating the effects of the concomitant transcranial direct current stimulation (tDCS) and computer-assisted cognitive training on episodic memory in individuals with TBI. This study may contribute to the development of evidence-based low-risk and low-cost rehabilitation treatments for TBI survivors with memory impairments. The topic is timely and clinically very relevant due to the lack of consensus and scarcity of evidence about the effects of cognitive training in addition to tDCS sessions in patients with TBI. The manuscript is well written and designed. However, I have several comments. Specific comments: Although the authors provided the power sample size analysis, it is unclear whether n=36 is the total number or number per group. If the provided is the total number, giving 12 patients per group, I'm not sure whether the sample size is large enough, considering age, post-TBI time, and TBI severity difference. How the possible gender and TBI severity difference will be valued? The minimal post-TBI period is 6 months. Did the author pre-set the maximal post-TBI period as an exclusion criterion? The first sentence of the Design subsection of the Methods and analysis section – the first sentence has the repeated word “study”.
---

VERSION 1 – AUTHOR RESPONSE

Dear Dr. Shih-kuen Cheng,

First of all, we would like to thank you for your insightful and highly relevant questions, which have already helped us improve the quality of our manuscript. We did our best to answer all of them as thoroughly as possible, however, in case our answers do not suffice, please let us know and we will be more than willing to resolve any further doubts.

1) The cognitive function this study wanted to focus is episodic memory. As also noted by the author, episodic memory is related to the memory for the spatial and temporal context of experienced events. How is this nature of episodic memory (spatial and temporal context) indexed by the RAVLT? More specifically, what scores obtained in the RAVLT would be used to index episodic memory? Previous studies, such as Lesniak et al. (2014)¹, have adopted the Rey's Auditory Verbal Learning Test as a measure of episodic memory. According to the Rey's Auditory Verbal Learning Test handbook (Brazilian Version), the RAVLT is a test conceived to “assess episodic memory processes

through word list repetition". The RAVLT handbook also mentions that tests in which content is presented and later required to be recalled - such as those that involve learning word lists - are thought to assess episodic declarative memory². Along the same lines, the Compendium of Neuropsychological Tests states that "episodic memory tests can also be differentiated according to the degree of inherent organization of the material to be memorized and the mode of reproduction (Spaan et al., 2003). Tasks that require recall of semantically unrelated material (e.g., an unrelated list of words such as the RAVLT or Buschke SRT) are thought to be more difficult because they require more effortful strategies for encoding and retrieval than do story recall tasks (e.g., WMS-III Logical Memory) or semantically related word lists (e.g., CVLT-II, HVLt-R)"³. In this context, the RAVLT provides measures for both short and long-term episodic memory. Short-term episodic memory, for example, may be evaluated by the scores obtained for lists A1, B1 and A6 whereas long-term episodic memory may be evaluated by the scores obtained for list A7 and the Recognition Test.² We have decided to use the score obtained for the A7 list, because in this condition the patient is required to retrieve the first list of words (A1) without any cues 20 minutes after his/her most recent A1 list repetition (A6), thus enabling us to assess coding, long-term retention and retrieval of episodic information.

References

- 1 Lesniak M, Polanowska K, Seniow J, Czlonkowska A. Effects of Repeated Anodal tDCS Coupled With Cognitive Training for Patients With Severe Traumatic Brain Injury: A Pilot Randomized Controlled Trial. *Journal of Head Trauma Rehabilitation*. 2014;29(3):E20-E9.
- 2 De Paula J, Fernandes Malloy-Diniz L. *Teste de Aprendizagem Auditivo-Verbal de Rey (RAVLT)*. São Paulo: Vetor Editora Psico-Pedagógica Ltda. 2018.
- 3 Strauss, E., Sherman, E. M. S. & Spreen, O. *A compendium of neuropsychological tests. Administration, Norms, and Commentary*. Third edition, Oxford University Press, New York, NY, 2006.

2) As also noted by the authors in the introduction, episodic memory is subserved by the hippocampus and the left DLPFC (along with a number of other brain regions). It is therefore understandable that the left DLPFC is chosen as the location for the tDCS, but why was the bilateral temporal cortex also selected as the target sites? It is also not clear why it is predicted that stimulating the BTC would have a greater episodic memory enhancement than stimulating the left DLPFC? If indeed such difference is observed, would that a result of different stimulation magnitude over the DLPFC and the hippocampus?

In addition to the left DLPFC, we have also selected the temporal cortex as a stimulation site because previous studies have applied tDCS stimulation over temporal regions – such as T3 and T4 - to investigate memory effects, albeit with mixed results and different populations (Stroke and Alzheimer patients, for example)^{1 2 3}. Furthermore, there is a memory system associated with the medial temporal lobe of which the hippocampus is a part and the hippocampal neuronal firing patterns are thought to reflect "unique conjunctions of stimuli with their significance, the animal's specific behaviors, and the places and contexts in which the stimuli occur"⁴. We have noticed that these studies were not included in our manuscript and, because of that, a sentence was added in the "Intervention" subsection in order to justify the use of this particular montage: "T3, T4 and F3 regions have been chosen for this protocol because other studies have investigated the effects of tDCS on

memory by placing the electrodes over temporal and left dorsolateral prefrontal regions". Nonetheless, it must be stressed that, to the best of our knowledge, no other study has used the bilateral temporal cortex anodal stimulation to investigate episodic memory in TBI patients. As for the assumption that BTC stimulation would have a greater episodic memory enhancement than IDLPFC stimulation, we have decided that it would more prudent not to assume such difference as certain and we have changed the sentence in the Introduction that read "active tDCS over the BTC will demonstrate higher episodic memory scores compared to the IDLPFC" for the following sentence: "there might be significant score differences on episodic memory test between patients who were stimulated over the BTC and those stimulated over the IDPFC". If any difference is observed in episodic memory performance between the BTC and the IDLPFC groups, different stimulation magnitude is a valid hypothesis to account for such difference and will definitely be taken into account for the interpretation of this study results.

References

- 1 Ross LA, McCoy D, Coslett HB, Olson IR, Wolk DA. Improved proper name recall in aging after electrical stimulation of the anterior temporal lobes. *Front Aging Neurosci.* 2011;3:16.
- 2 Yun GJ, Chun MH, Kim BR. The Effects of Transcranial Direct-Current Stimulation on Cognition in Stroke Patients. *J Stroke.* 2015;17(3):354-358.
- 3 Boggio PS, Khoury LP, Martins DC, Martins OE, de Macedo EC, Fregni F. Temporal cortex direct current stimulation enhances performance on a visual recognition memory task in Alzheimer disease. *J Neurol Neurosurg Psychiatry.* 2009 Apr;80(4):444-7.
- 4 Eichenbaum H, Lipton PA. Towards a functional organization of the medial temporal lobe memory system: role of the parahippocampal and medial entorhinal cortical areas. *Hippocampus.* 2008;18(12):1314-24

3) The RAVLT will be tested three times in the protocol (Visit 1 as the baseline, visit 10, and visit 11). Will the same lists be used in the three tests?

For the first and the third assessments, which are three-months apart, the same lists will be used. However, a different list of words (alternate form) will be used for the second assessment to circumvent any possible practice effect, since the first and second assessments take place within a period that is shorter than three months (two weeks).

4) The protocol did not make it clear the temporal relationship between the tDCS stimulation and the cognitive training.

tDCS stimulation and cognitive training will be taking place simultaneously, that is, patients will be receiving either active or sham tDCS stimulation while performing the computer-based cognitive training. The words "concomitant" and "online" have been used in other studies investigating the effects of tDCS and cognitive training. However, those words might be misleading, in the sense that "online cognitive training" might be mistaken for a cognitive training that is to be completed through the internet (which is not the case) and "concomitant cognitive training" might be mistaken for a training that takes place immediately after the tDCS period (the definition of "concomitant" contributes

to that confusion: “a phenomenon that naturally accompanies or follows something”). Accordingly, to make our method clearer, we have decided to use the word “concurrent” before “cognitive training” when referring to our study in order to emphasize the simultaneous aspect of the tDCS stimulation and cognitive training. We are open to other suggestions.

5) The sample size was estimated based on the 3 (DLPFC, BTC, and sham stimulation) by 3 (testing times) interaction effect on the episodic memory performance with an effect size of .25. What was the basis for the effect size of .25?

Indeed, Cohen’s criteria considers effect sizes between 0.2 to 0.5 to be small, however, the effect size of 0.25 was chosen after such effect size was observed in the pilot study we carried out with 4 patients (all of them treated in an active stimulation condition).

Reply to Dr. Denis Bragin:

Dear Dr. Denis Bragin,

We would like to thank you for taking your time to assess the manuscript we have submitted to the BMJ and would like to thank you for your insightful questions that have already helped us improve the quality of our text. We have attempted to answer your questions as thoroughly as possible.

1) Although the authors provided the power sample size analysis, it is unclear whether $n=36$ is the total number or number per group. If the provided is the total number, giving 12 patients per group, I’m not sure whether the sample size is large enough, considering age, post-TBI time, and TBI severity difference.

The total number of patients is 36 and, therefore, there will be 12 patients per group. We have added the following sentence in the “Sample Size Calculation” section in order to make this information clearer: “12 patients per group”. Age, post-TBI time and TBI severity are not covariates in our study, however, in order to favor interpretation of results, our patients’ age range is small (between 18 yrs and 50 yrs), post-TBI time is limited (between 6 months and 18 months) and TBI severity is moderate-to-severe. In this context, we consider our group to be homogeneous enough to allow us to generalize the results of our intervention.

2) How the possible gender and TBI severity difference will be valued?

Gender/Sex and TBI severity will not be considered as covariates due to sample size limitations. In order to indicate that not considering Sex and TBI severity as covariates due to sample size restrictions is a limitation of our study, we have decided to include the following sentence in the “Strengths and Limitations” section of our manuscript: “Due to sample size restrictions, sex and TBI severity will not be considered as covariates, which might be a limitation of this study”. It is noteworthy that severe TBI and moderate TBI are considered as a single entity for investigation purposes in many studies, in part because of the permanent physical, cognitive and behavioral impairments that are observed in such patients in comparison to mild TBI patients.^{1 2} As for sex differences, a recent study aimed at characterizing the demographic, social and economic profile of TBI patients in Brazil showed that men were hospitalized almost 3.5 times more frequently for TBI than women and that the incidence of TBI in the male population was 102/100,000/year³.

References

1 Timmer ML, Jacobs B, Schonherr MC, Spikman JM, van der Naalt J. The Spectrum of Long-Term Behavioral Disturbances and Provided Care After Traumatic Brain Injury. *Front Neurol.* 2020 Apr 7;11:246

2 Azouvi P, Arnould A, Dromer E, Vallat-Azouvi C. Neuropsychology of traumatic brain injury: An expert overview. Rev Neurol (Paris). 2017 Jul-Aug;173(7-8):461-472

3 de Almeida CE, de Sousa Filho JL, Dourado JC, Gontijo PA, Dellaretti MA, Costa BS. Traumatic Brain Injury Epidemiology in Brazil. World Neurosurg. 2016 Mar;87:540-7

3) The minimal post-TBI period is 6 months. Did the author pre-set the maximal post-TBI period as an exclusion criterion?

We did set a maximal post-TBI period as an exclusion criterion, which is 18 months. This information was not in our manuscript and we have added it to the Exclusion Criteria subsection, as follows: "Time after trauma > 18 months".

4) The first sentence of the Design subsection of the Methods and analysis section – the first sentence has the repeated word "study".

R: As suggested, we have removed the word "study" at the beginning of the sentence to eliminate the aforementioned repetition.

VERSION 2 – REVIEW

REVIEWER	Bragin, Denis University of New Mexico School of Medicine, Neurosurgery
REVIEW RETURNED	21-Apr-2021
GENERAL COMMENTS	No further concerns.